# A large-scale proteomics resource of circulating extracellular vesicles for biomarker discovery in pancreatic cancer

Bruno Bockorny[1,2†], Lakshmi Muthuswamy[3†], Ling Huang[4†], Marco Hadisurya[5], Christine Maria Lim[6], Leo L Tsai[2,7], Ritu R Gill[2,7], Jesse L Wei[2,7], Andrea J Bullock[1,2], Joseph E Grossman[8], Robert J Besaw[1], Supraja Narasimhan[9], Weiguo Andy Tao[5], Sofia Perea[1], Mandeep S Sawhney[2,10], Steven D Freedman[2,10], Manuel Hildago[11,12†], Anton Iliuk[13*†], Senthil K Muthuswamy[14*†]

[1]Division of Medical Oncology, Beth Israel Deaconess Medical Center, Boston, United States; [2]Harvard Medical School, Boston, United States; [3]Blueprint Medicines, Cambridge, United States; [4]Henry Ford Cancer Institute, Detroit, United States; [5]Department of Biochemistry, Purdue University West Lafayette, West Lafayette, United States; [6]Nanyang Technological University, Singapore, Singapore; [7]Department of Radiology, Beth Israel Deaconess Medical Center, Boston, United States; [8]Agenus Inc, Lexington, United States; [9]Deciphera Pharmaceuticals, Waltham, United States; [10]Division of Gastroenterology, Beth Israel Deaconess Medical Center, Boston, United States; [11]Division of Hematology-Oncology, Weill Cornell Medical College, New York, United States; [12]New York-Presbyterian Hospital, New York, United States; [13]Tymora Analytical Operations, West Lafayette, United States; [14]National Cancer Institute, Bethesda, United States

*For correspondence:
anton.iliuk@tymora-analytical.com (AI);
senthil.muthuswamy@nih.gov (SKM)

†These authors contributed equally to this work

## eLife Assessment

The authors analyze a comprehensive cohort of human plasma samples to identify an extracellular vesicles protein signature for early diagnosis of pancreatic cancer. The application of liquid biopsies is **valuable**, and the work addresses a key clinical problem as pancreas cancer is often diagnosed in late stages. The strength of evidence is **solid**. Altogether, this work supports the potential use of extracellular vesicles in clinical settings, with promising value to scientists and clinicians.

**Abstract** Pancreatic cancer has the worst prognosis of all common tumors. Earlier cancer diagnosis could increase survival rates and better assessment of metastatic disease could improve patient care. As such, there is an urgent need to develop biomarkers to diagnose this deadly malignancy. Analyzing circulating extracellular vesicles (cEVs) using 'liquid biopsies' offers an attractive approach to diagnose and monitor disease status. However, it is important to differentiate EV-associated proteins enriched in patients with pancreatic ductal adenocarcinoma (PDAC) from those with benign pancreatic diseases such as chronic pancreatitis and intraductal papillary mucinous neoplasm (IPMN). To meet this need, we combined the novel EVtrap method for highly efficient isolation of EVs from plasma and conducted proteomics analysis of samples from 124 individuals, including patients with PDAC, benign pancreatic diseases and controls. On average, 912 EV proteins were identified per 100 µL of plasma. EVs containing high levels of PDCD6IP, SERPINA12, and RUVBL2 were associated with PDAC compared to the benign diseases in both discovery and validation cohorts. EVs with PSMB4, RUVBL2, and ANKAR were associated with metastasis, and those with CRP, RALB, and CD55 correlated with poor clinical prognosis. Finally, we validated a seven EV

protein PDAC signature against a background of benign pancreatic diseases that yielded an 89% prediction accuracy for the diagnosis of PDAC. To our knowledge, our study represents the largest proteomics profiling of circulating EVs ever conducted in pancreatic cancer and provides a valuable open-source atlas to the scientific community with a comprehensive catalogue of novel cEVs that may assist in the development of biomarkers and improve the outcomes of patients with PDAC.

## Introduction

Pancreatic ductal adenocarcinoma (PDAC) has the worst prognosis of all common tumors, with a 5-year survival of 10% (*Siegel et al., 2020*). With rising incidence, it is expected that PDAC will become the second leading cause of cancer-related deaths by 2030 (*Rahib et al., 2014*). A critical factor for this dismal development is the late diagnosis, with less than 20% of patients presenting with a potentially resectable and curable tumor (*Kamisawa et al., 2016*; *Rahib et al., 2016*; *Bockorny et al., 2022*). Earlier cancer diagnosis could increase the survival rates by an estimated fivefold, and more reliable and real-time assessment of treatment effects in patients with cancer could improve quality of life and reduce healthcare costs (*Ghatnekar et al., 2013*; *Matsuno et al., 2004*). Unfortunately, there are no credentialed serologic biomarkers with high enough performance to assist in the early detection of PDAC. The best-established biomarker for PDAC, carbohydrate antigen 19–9 (CA19-9), is fraught with poor sensitivity and specificity and is only used for monitoring disease on treatment or after surgical resection (*Locker et al., 2006*; *Galli et al., 2013*).

Extracellular vesicles (EVs), including exosomes and microvesicles, are nanosized particles released by most cell types and can be detected in the circulation (*Chen et al., 2017*). EVs play important roles in transmission of oncogenic and inflammatory signals (*Costa-Silva et al., 2015*), communications between cells and their microenvironment (*van Niel et al., 2022*). In addition, exoDNA, exoRNA and protein profiles highly reflect parental cells, therefore offering an attractive strategy for diagnosing cancers non-invasively by analyzing EVs in the circulation (*Costa-Silva et al., 2015*; *Melo et al., 2015*). Previous studies employed EVs to discover biomarkers for PDAC (*Melo et al., 2015*; *Madhavan et al., 2015*; *Yang et al., 2017*; *Castillo et al., 2018*); however, those discovery proteomics experiments were carried out using cell lines or tumor tissue, which are not representative of the heterogeneity of human PDAC and are unable to recapitulate the systemic responses to cancer (*Madhavan et al., 2015*; *Yang et al., 2017*; *Castillo et al., 2018*). In addition, the EV biomarkers discovered in those studies have been compared only against healthy controls (*Madhavan et al., 2015*; *Yang et al., 2017*; *Castillo et al., 2018*). It is unclear how they would perform in subjects with underlying benign diseases of the pancreas, which is highly desirable from the clinical standpoint as many patients with PDAC have underlying chronic pancreatitis and cysts.

To meet this need, we conducted a large EV proteomics study from peripheral blood across a range of patients with pancreatic cancer, benign pancreatic diseases such as chronic pancreatitis and intraductal papillary mucinous neoplasm (IPMN), and healthy controls. Circulating EV (cEV) proteins detected included those involved in metabolism and immune regulation, in addition to proteins involved in protein binding, exocytosis, endocytosis and regulation of cellular protein localization that have been identified in previous studies (*Fahrmann et al., 2020*; *Hoshino et al., 2020*). We subsequently discovered multiple biomarker candidates for cancer diagnosis and verified several of them in an independent cohort of patients with the potential to aid in diagnosing pancreatic cancer. In addition, we identified a set of cEV proteins associated with metastasis which could provide a valuable resource for future biomarker studies.

## Results

### Proteomics characterization of circulating EVs

In this study, we sought to identify proteins in extracellular vesicles in the blood that may be used as biomarkers for the diagnosis and prognosis of pancreatic cancer. With the approval of our institutional review board (DF/HCC IRB#17–640), we enrolled a total of 124 patients to the discovery cohort of this biomarker study (Methods and *Supplementary file 1*). Subjects in the pancreatic cancer group (N=93) had a mean age of 66.5 years (range, 37–91), and 48.4% were female. All subjects had biopsy-proven disease. Thirty subjects had early-stage disease (stages I-II) and 63 had advanced disease (stages

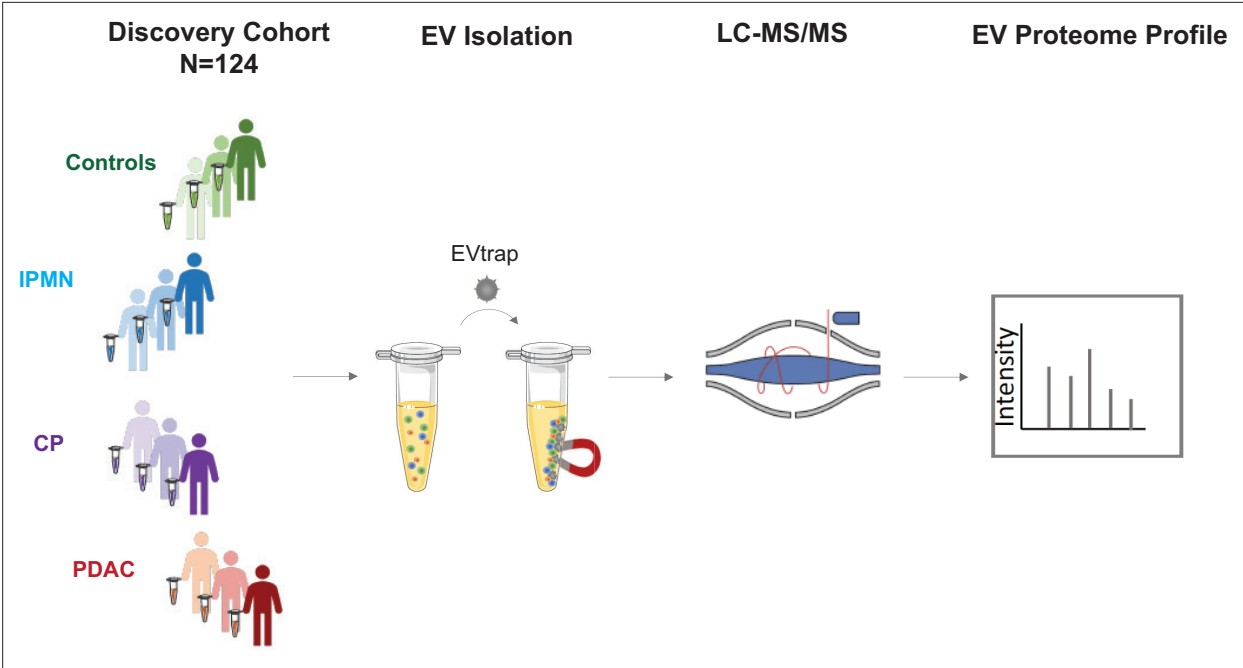

**Figure 1.** Study design. The discovery cohort was comprised of 124 individuals, including pancreatic ductal adenocarcinoma (PDAC, N=93), chronic pancreatitis (CP, N=12), intraductal papillary mucinous neoplasm (IPMN, N=8) and healthy controls (N=11). Plasma samples were processed for EV isolation using EVtrap and analyzed by liquid chromatography-tandem mass spectrometry (LC-MS/MS).

The online version of this article includes the following figure supplement(s) for figure 1:

**Figure supplement 1.** EVtrap isolation of extracellular vesicles.

**Figure supplement 2.** EV proteomics analytical performance.

III-IV). Patients with benign pancreatic diseases included chronic pancreatitis (N=12) with a mean age of 57.5 years (range 37–78) and with 50% females, whereas IPMN included individuals with main duct and side branch IPMNs (N=8) with a mean age of 68.2 years (range 50–89) and with 87.5% being females. Subjects in the healthy control group (N=11) had a mean age of 53.4 years (range, 31–83) with 54.5% females (*Supplementary file 1*).

We employed the novel EVtrap method (Extracellular Vesicles Total Recovery And Purification) to capture EVs from plasma samples and overcome the traditional laborious techniques for EV isolation, which are not scalable for large clinical studies. As described in recent reports, EVtrap is a magnetic bead-based isolation method that enables highly efficient capture of EVs from biofluids, confirmed by multiple common EV markers (*Iliuk et al., 2020*; *Wu et al., 2018*; *Nunez Lopez et al., 2022*; *Hinzman et al., 2022a*; *Hinzman et al., 2022b*). Previous analyses using electron microscopy and nanoparticle tracking also confirmed that the vast majority of particles isolated by EVtrap had diameters between 100 and 200 nm, consistent with exosomes (*Iliuk et al., 2020*). In addition, EVtrap isolates demonstrates higher abundance of CD9, a common exosome marker, as compared to isolates from other traditional EV isolation methods such as size exclusion chromatography and ultracentrifugation (*Iliuk et al., 2020*). Over 95% recovery yield can be achieved by EVtrap with less contamination from soluble proteins, a significant improvement over current commercially available methods as well as ultracentrifugation (*Iliuk et al., 2020*; *Nunez Lopez et al., 2022*; *Shuen et al., 2022*).

Following EV isolation, samples were digested in-solution and analyzed by liquid chromatography-tandem mass spectrometry (nanoLC-MS/MS) on a high-resolution mass spectrometer (Q-Exactive HF-X). The workflow for cEVs isolation and enrichment and subsequent cEV mass spectrometry analysis is illustrated in *Figure 1A*.

First, to confirm that EVtrap can efficiently isolate extracellular vesicles from plasma, a test plasma sample was processed to remove platelets and other large particles and enriched for EVs using EVtrap beads (see Methods for details). Transmission electron microscopy (TEM) analysis of the EV pellet showed cup-shaped extracellular vesicles (exosomes and microvesicles; *Figure 1—figure*

*supplement 1A*), and nanoparticle tracking analysis (NTA) using ZetaView instrument (Particle Metrix) demonstrated that the isolated EVs were in the 100–200 nm diameter range, with a mean diameter of 152 nm (*Figure 1—figure supplement 1B*). Second, to assess the technical reproducibility of the EV proteomics approach, the test plasma sample was processed in six replicates and Pearson correlation analysis revealed a very high correlation (r2 >0.97) between replicates (*Figure 1—figure supplement 2A*, *Supplementary file 2*). These results provided the confidence to proceed with the analysis of our discovery set of plasma from 124 subjects. In this cohort, we identified 1708 unique proteins (*Supplementary file 3*). The number of unique EV proteins detected per 100 µL of plasma sample varied from 817 to 1128, with an average of 912 unique proteins per sample (*Figure 1—figure supplement 2B*). We did not observe differences between non-tumor and tumor samples regarding the overall number of EV proteins identified. Within the PDAC group, we did not observe significant differences in the average number of EV proteins detected for different disease stages. Collectively, these data demonstrate high reproducibility of EV isolation and robust label-free MS quantification of cEVs.

## Diseases of the pancreas express distinct circulating EV proteome compared to controls

Next, we aimed at identifying specific cEV proteins associated with clinical parameters with the potential to serve as diagnostic biomarkers. We first compared the proteomics profile of individuals with underlying pancreatic diseases (PDAC, chronic pancreatitis and IPMN) against healthy controls. We selected EV proteins expressed in at least 50% of subjects in the disease group with a fold change of expression ≥2 or≤2 compared to controls and p-value ≤0.01 after adjusting for multiple testing. A total of 207 proteins were identified that met the criteria, with the largest number of differentially expressed markers in PDAC (176), followed by chronic pancreatitis (55) and IPMN (3) (*Supplementary file 4*). Principal component analysis (PCA) of these markers showed control samples as a tight cluster segregated away from PDAC samples but closer to IPMN and chronic pancreatitis patients (*Figure 2A*).

## Circulating EV proteome discriminates pancreatic cancer from benign pancreatic diseases

To further assess the potential of cEV proteins for cancer detection, we compared proteomic profiles of cEVs between patients with PDAC with those with underlying benign diseases of the pancreas (chronic pancreatitis and IPMN). We identified 182 differentially expressed proteins in malignant cases (92 over-expressed and 90 with reduced expression; *Supplementary file 5*). Several of those markers had remarkable overexpression in PDAC (greater than tenfold), including PDCD6IP, SERPINA12, RUVBL2, among others, as shown in the volcano plot (*Figure 2B*). Unsupervised clustering showed a clear separation between PDAC and benign pancreatic diseases. Individuals with IPMN were more closely related to controls, whereas chronic pancreatitis cases were more related to PDAC (*Figure 2C*). In addition, the PDAC cohort was separated into two subgroups: the first, enriched for early-stage tumors and more closely related to the other pancreatic diseases (chronic pancreatitis and IPMN); the second, enriched for advanced and metastatic cases with expression profiles further apart from early-stage cancer and pancreatic diseases (*Figure 2C*). We further noticed that some proteins such as PDCD6IP, SERPINA12, KRT20 showed statistically significant population-wise enrichment in pancreatic cancer compared to benign pancreatic diseases (*Figure 2D*, *Figure 2—figure supplement 1*). Together, these data indicate the existence of EV markers that can separate controls, benign and malignant pancreatic diseases, as well as proteins that separate early versus late-stage PDAC, suggesting their potential to serve as diagnostic biomarkers.

## Functional and systems biology of cEV proteome

To gain molecular insight into the functions of the 182 proteins differentially expressed in pancreatic cancer as compared to benign pancreatic diseases, we conducted pathway analysis using the Gene Ontology (GO) and REACTOME databases (*Supplementary file 5*). We identified protein modules in protein localization, biomolecule binding/docking, peptidase activities among changes enriched in PDAC compared to benign diseases (*Figure 2—figure supplement 2*). Interestingly, KRT20 (keratin 20), a gastrointestinal epithelia-associated keratin, was increased in PDAC patient EVs, while keratins associated basal cells, KRT4, KRT15, and KRT3, were reduced. KRT20 overexpression is frequently

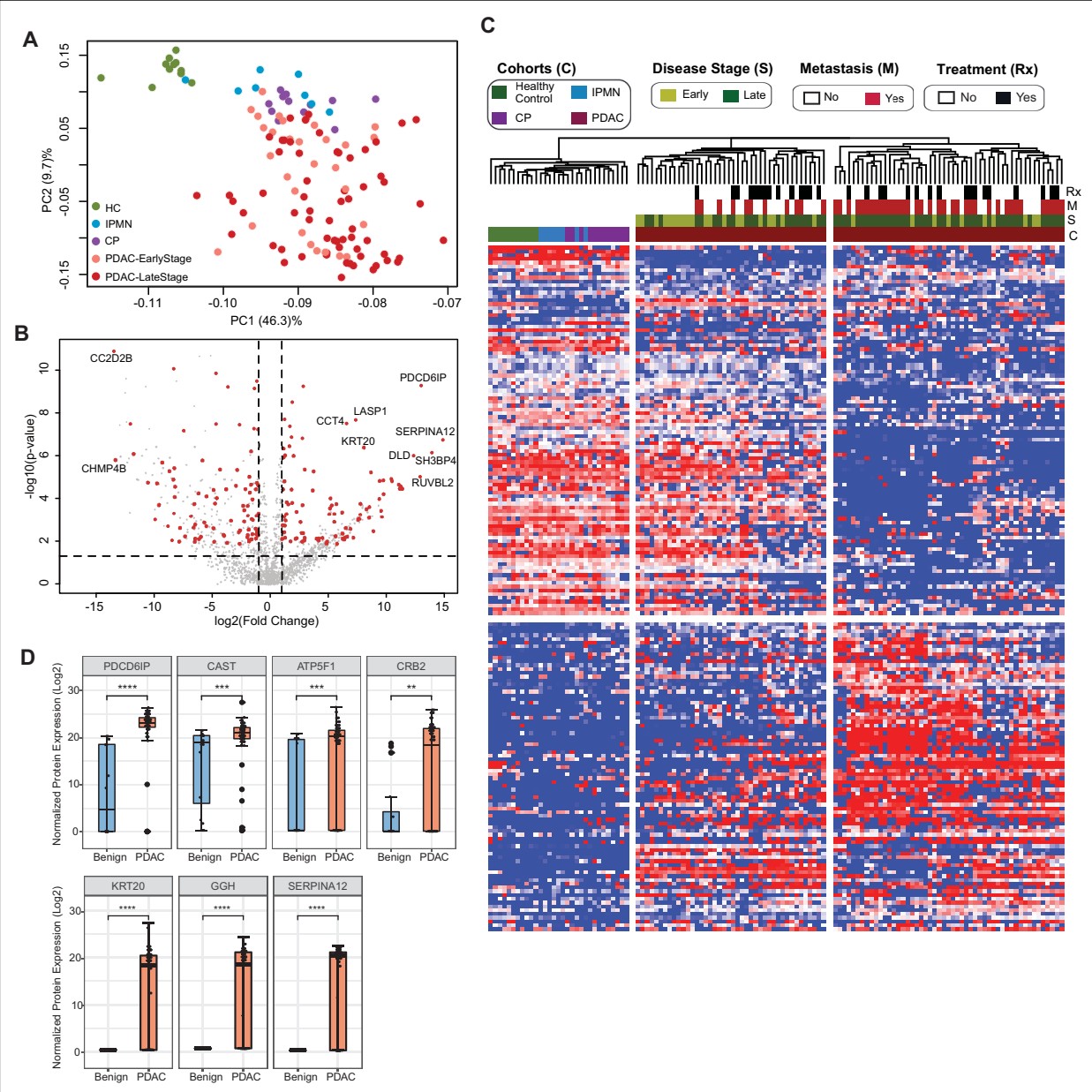

**Figure 2.** Identification of cEV proteins differentially expressed in disease groups. (**A**) Principal component analysis of cEV proteins differentially expressed in the plasma of patients with pancreatic diseases compared to controls. Each dot indicates one individual enrolled in the study: green, controls; blue, patients with intraductal papillary mucinous neoplasm (IPMN); purple, patients with chronic pancreatitis (CP); salmon, early stage (stages I and II) pancreatic ductal adenocarcinoma (PDAC); red, late stage (stages III and IV) PDAC. (**B**) Volcano plot of circulating EV proteins enriched in the plasma of patients with PDAC versus benign pancreatic diseases. X-axis, log base 2 of fold changes; Y-axis, negative of the log base 10 of p values. (**C**) Heatmap of cEV proteins differentially expressed in the plasma of patients with pancreatic diseases compared to controls. Designations of clinical parameters were indicated at the top of the heatmap. (**D**) Expression of enriched cEV proteins in patients with PDAC (N=93) versus benign pancreatic diseases (N=20). Each dot indicates the target protein signal from one patient. Y-axis, normalized log base 2 of protein signals detected by mass spectrometry; Error bars, min and max values; lines in boxes, median values. * p≤0.05, ** p≤0.01, *** p≤0.001, **** p≤0.0001.

The online version of this article includes the following figure supplement(s) for figure 2:

**Figure supplement 1.** Heatmap of abundance of 25 proteins enriched and 25 proteins reduced in EVs from PDAC patients compared to EVs from patients without cancer.

**Figure supplement 2.** Network analyses of cEV proteins differentially expressed in PDAC compared to benign pancreatic diseases.

found in pancreatic tumor tissues and correlates with poor prognosis (*Schmitz-Winnenthal et al., 2006*), suggesting a biological basis for their high levels in the cEVs of PDAC patients.

Interestingly, proteins associated with immunological functions showed complex regulation with increased representation of leukocyte mediated immunity (GO:0002443), leukocyte degranulation (GO:0043299), myeloid leukocyte activation (GO:0002274), and decrease in Fc receptor signaling (GO: 0038093), regulation of complement activation (GO:0030449), and immune effector process (GO:0002252; *Supplementary file 5*). These data suggest that direct profiling of cEVs from patient plasma provided unique insights into systemic changes in immune biology during pancreatic cancer development, which is lacked in analysis restricted to tissue or cell models.

## Circulating EV proteomics reveal markers associated with metastasis and worse prognosis

We then investigated whether cEV proteins can assist in the distinction of metastatic versus non-metastatic pancreatic cancer. We compared the cEV proteome profiles of individuals with metastatic cancer to those without metastasis and identified 85 proteins differentially expressed between the two groups (*Supplementary file 6*). Supervised clustering between metastatic and non-metastatic diseases showed a clear separation with two distinct expression patterns (*Figure 3A*). In particular, PSMB4, RUVBL2, and ANKAR (*Figure 3B*) EV protein levels were increased in patients with metastatic disease, whereas RAP2B, SERPINA12, and IGLV4-69 abundance levels were decreased in the cEVs of patients with metastasis (*Figure 3C*). Together, these findings suggest the presence of a core set of cEV proteins with the potential to distinguish early versus metastatic pancreatic cancer.

We further analyzed whether the expression of certain cEV proteins had prognostic relevance in our cohort. We first classified individuals with PDAC as having low or high expression of any given markers based on each marker's first and third quartile. Survival was estimated by the Kaplan Meier method. We identified that the cEV expression of RALB, CRP, and CD55 had a significant correlation with overall survival, with a trend for PDCD6IP (*Figure 3D*).

## Validation of cEV markers using parallel reaction monitoring and identification an EV protein signature for pancreatic cancer diagnosis

Because pancreatic cancer is extremely heterogeneous, the chance of identifying a single biomarker with sufficient diagnostic performance is likely low. Instead, the identification of a panel of candidate markers may have enhanced diagnostic performance.

To identify a signature that shows the most discriminatory power between 'benign diseases' and 'PDAC,' we employed a binary classification approach using Support Vector Machines (SVM). Classification models, built based on a large number of proteins, contain irrelevant markers that can reduce the predictive accuracy. Hence, we implemented a consensus feature selection method based on two algorithms: one using recursive feature elimination (RFE) algorithm (SVM-RFE; *Guyon et al., 2002*) and second, RFE combined with a non-parametric Wilcox rank test (sigFeature; *Das et al., 2020*). The top 16 markers were selected whose classification performance can be tested in the independent validation cohort (*Supplementary file 7*). A summary of selection process is shown in *Figure 4—figure supplement 1*. The classification performance of these 16 markers, individual and in all combinations, were tested using 80% training data and evaluated in the remaining 20% test data. The quality of training was assessed using five repetitions of tenfold cross-validation. The optimal kernel parameters were estimated by tuning over a wide range of values. Receiver operating characteristic (ROC) analysis was used as the metric to assess the performance of the classifier model. We found a set of seven EV protein signature comprised of RUVBL2, PDCD6IP, ATP5F1, DLD, KRT20, CCT4, and SERPINAI2, that gave 100% accuracy when tested in the discovery cohort (*Figure 4—figure supplement 2*, *Figure 4—figure supplement 3*). Recurrence of these putative markers in our dataset varied from 55% to 97%.

The model was further validated on an independent validation cohort whose proteome was obtained using an alternate technology, parallel reaction monitoring (PRM) mass spectrometry. The markers chosen for validation included 16 markers selected for SVM classification model and an additional 9 markers to result in top 25 markers that are significantly differentially expressed in the discovery cohort with a fold change increase in PDAC ≥5.5 and p-value ≤0.01 (Methods, *Figure 4—figure supplement 4*). The independent validation cohort consisted of 36 new subjects (24 with PDAC, 6

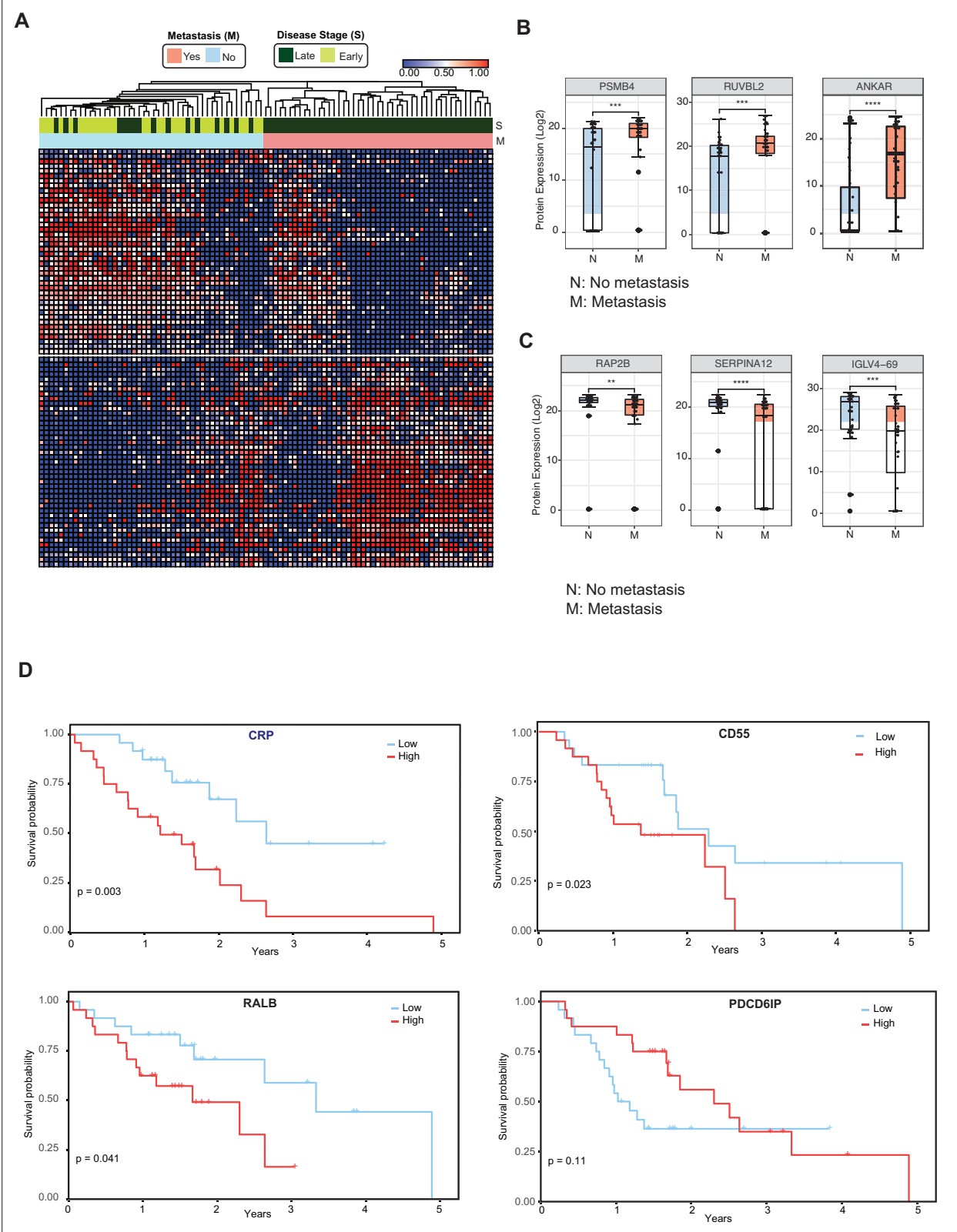

**Figure 3.** Circulating EV proteomics reveal markers associated with metastasis and worse prognosis. (**A**) Heatmap showing EV proteins differentially expressed in the plasma of metastatic versus non-metastatic PDAC. Designations of clinical parameters are indicated at the top of the heatmap. (**B**) Expression patterns of cEV proteins associated with metastatic disease. Y-axis, normalized log base 2 of protein signals detected by mass spectrometry; N, non-metastatic PDAC group (N=46); M, metastatic PDAC group (N=47). Each dot indicated the target protein signal from one patient. Error bars,

*Figure 3 continued on next page*

*Figure 3 continued*

min and max values; lines in boxes, median values. * p≤0.05, ** p≤0.01, *** p≤0.001, **** p≤0.0001. (**C**) As is (**B**), except for cEV markers with increased expression in non-metastatic PDAC. (**D**) Correlation of cEV marker expression with survival. Kaplan–Meier curves and log-rank test p values of representative survival cEV markers quantified in the discovery cohort.

with chronic pancreatitis, and 6 with IPMN; *Supplementary file 9*). A total of 10 proteins, including all 7 signature proteins, showed a significant difference (p<0.05) in patients with PDAC as compared to benign pancreatic diseases (*Figure 4A*). The performance of individual validated markers according to the specific underlying disease in the validation cohort is presented in *Figure 4—figure supplement 3*. The performance of seven EV protein signature was further tested using SVM model, in our independent validation cohort, yielding an 89% prediction accuracy (*Figure 4B*, *Figure 4—figure supplement 4*). As expected, we observed that no single marker achieved sufficiently high sensitivity and specificity as the combined model for the diagnosis of pancreatic cancer.

## Discussion

Extracellular vesicles hold a great promise as a source of potential biomarkers, making them attractive candidates for liquid biopsy tests. Previously, we reported that organoid cultures of pancreatic cancer could serve as models to discover tissue-derived EV proteins with high specificity for PDAC, as opposed to chronic pancreatitis and other benign gastrointestinal diseases (*Huang et al., 2020*). A shortcoming of these tissue-based studies is the inability to discover markers associated to the systemic responses to cancer. Here, we performed a large-scale, comprehensive analysis of circulating EV proteomes directly from plasma samples of 124 patients, with subsequent validation in a separate cohort of 36 patients. To our knowledge, this represents the largest proteomics profiling dataset of circulating EVs conducted in pancreatic cancer to date. In this study, we identified and validated new EV markers from plasma that distinguish patients with pancreatic cancer from subjects with benign pancreatic diseases. Furthermore, we discovered several cEV proteins associated with metastatic disease and poor prognosis. In contrast to the prior studies of experimental cell models or tissues extracts that were examined only against healthy subjects (*Madhavan et al., 2015*; *Yang et al., 2017*; *Castillo et al., 2018*), we report the identification of EV proteins in plasma of patients with pancreatic cancer compared to patients with underlying pancreatic diseases, which is clinically relevant as many patients with pancreatic cancer have underlying chronic inflammation and premalignant cystic lesions.

In addition, our study demonstrated the feasibility of using the novel EVtrap method *Iliuk et al., 2020*; *Wu et al., 2018* for discovery of hundreds of EV proteins directly from a small volume (100 µL) of plasma samples. This methodological advance can be adopted for biomarker discovery in other cancer types. Other workflows traditionally employed for EV isolation from blood samples require laborious techniques including lengthy ultracentrifugation steps which are unsuitable for large-scale studies (*LeBleu and Kalluri, 2020*).

We identified several EV proteins as significantly associated with metastasis or survival. For instance, PSMB4 and RUVBL2 levels were increased in cEVs of patients with metastatic PDAC. Notably, PSMB4 (proteasome subunit beta type-4), a protein of the ubiquitin-proteasome degradation pathway, has been identified as the first proteasomal subunit with oncogenic properties and associated to poor prognosis in several tumors including melanoma, breast, and ovarian cancers (*Liu et al., 2016*; *Zhang et al., 2017*; *Zheng et al., 2015*; *Lee et al., 2014*). As expected, the EV proteomic profiles of PDAC patients exhibited significant heterogeneity. While the above-mentioned markers exhibited strong association with disease states at population levels, their abundances in individual patients varied significantly. Those observations highlight the need to develop multi-protein panels for pancreatic cancer diagnosis and prognosis.

We also discovered RALB, CRP, and CD55 expression on EVs to have a significant correlation with poor survival, while PDCD6IP expression was associated with improved outcomes. Interestingly, PDCD6IP (programmed cell death 6-interacting protein), was also identified as a PDAC-enriched protein in the tissue-based proteomics studies from *Le Large et al., 2020* and *Hoshino et al., 2020*. In line with our findings, its tissue expression in liver metastasis of pancreatic cancer has been found to also correlate with improved prognosis in patients with PDAC in the study of *Law et al., 2020*. Collectively, these data suggest that some tissue-specific proteins can be isolated from circulating EVs

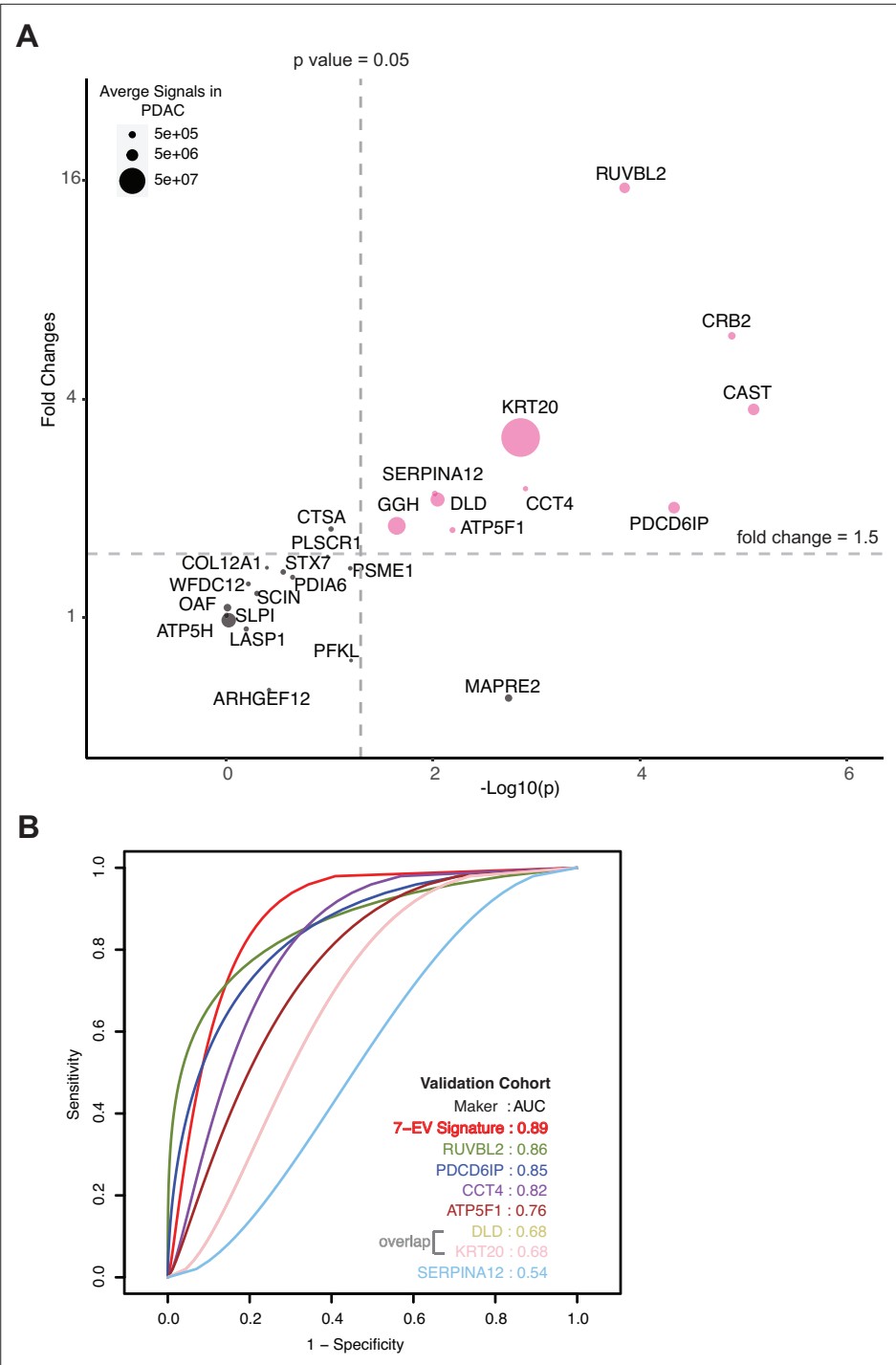

**Figure 4.** Validation of cEV markers and identification of seven EV protein signature for PDAC diagnosis. (**A**) Differences of cEV protein abundances between patients with PDAC (n=24) and benign pancreatic diseases (chronic pancreatitis and IPMN) (n=12). x axis, minus log p values of protein abundance differences between PDAC and benign groups; y axis, average fold changes of proteins in PDAC group compared to benign group. Size of bubbles indicate average protein abundances in PDAC group. Pink color, proteins that had at least twofold enrichment in PDAC group (p<0.05). (**B**) ROC curves were calculated for individual cEV markers as well as for the seven EV protein PDAC signature combination to determine optimum diagnostic performance.

The online version of this article includes the following figure supplement(s) for figure 4:

**Figure supplement 1.** Summary of selection process to develop EV signature for pancreatic cancer diagnosis.

*Figure 4 continued on next page*

*Figure 4 continued*

**Figure supplement 2.** Diagnostic performance of the seven EV protein signature compared to performance of each of the seven individual marker in patients with benign pancreatic diseases (N=12) or PDAC (N=24).

**Figure supplement 3.** Validation of individual cEV proteins in an independent cohort of patients.

**Figure supplement 4.** Performance of PDAC EV signature in both discovery (benign =20, PDAC=93) and validation (benign =12, PDAC=24) cohorts.

and their quantifiable levels in the blood may have the potential to serve as diagnostic or prognostic biomarkers in pancreatic cancer.

In our validation studies, all seven putative markers identified from the model were significantly enriched in the plasma of PDAC patients. Based on the top seven markers, we derived a seven EV protein panel that yielded an 89% prediction accuracy for diagnosing pancreatic cancer. A recent modeling study showed that a new diagnostic assay for PDAC would have to perform with a minimum sensitivity of 88% and a specificity of 85% to reduce healthcare expenditure and prolong survival (*Ghatnekar et al., 2013*). Serum CA19-9, the best-established blood test for PDAC, has a pooled sensitivity of 75.4% and a specificity of 77.6% (*Zhang et al., 2015*). It commonly rises late in the disease and may be elevated in nonmalignant conditions such as biliary obstruction and pancreatitis, making it unsuitable as a diagnostic biomarker for PDAC (*Duffy et al., 2010*). As such, our seven EV protein signature with 89% prediction accuracy serves as a proof-of-concept and has the potential to facilitate the further development of biomarker tests for pancreatic cancer. We anticipate that for clinical use application, an even higher diagnostic performance is needed. Future studies are warranted to investigate if combining our validated cEV proteins with other biomarkers such as cell free DNA, serum proteins or metabolites, as a multi-analyte biomarker assay, would yield higher accuracy in diagnosing pancreatic cancer.

While our work involved a large cohort with 160 patients, the single-center nature is an inherent limitation of our study. Also, it would be ideal to perform validation with a larger cohort of controls to achieve greater statistical power. To balance this limitation, we increased the rigor of our validation by selecting controls with underlying benign diseases of pancreas as opposed to healthy volunteers, and an alternate quantitative technology for measuring protein abundance (Parallel Reaction Monitoring Mass Spectrometry) instead of MS-LC. While this approach increased the generalizability, it marginally reduced model prediction. Thus, the performance of our seven EV protein PDAC panel should be cross-validated in larger and multicenter populations. In addition, in this work we only used EVtrap as EV isolation method and mass spectrometry for protein quantification, and it is possible that there was some degree of heterogeneity in the extracellular vesicles analyzed. The clinical impact of biomarkers identified in our study will need to be cross-validated using other methods.

With no major treatment breakthrough for pancreatic cancer in the last decade, every effort should be made to diagnose this deadly cancer at earlier stages and to discover new proteins involved in tumorigenesis. Our study provides a valuable open resource to the scientific community with a comprehensive catalog of novel proteins packaged inside circulating EVs that may assist in the development of novel biomarkers and improve the outcomes of patients with pancreatic cancer.

## Methods
### Study design and patient demographics

We conducted this study at Beth Israel Deaconess Medical Center with the approval of the Harvard Cancer Center Institutional Review Board (IRB#17–640). All subjects provided written informed consent. Clinical data and blood samples were prospectively collected from 2017 to 2019 from patients with pancreatic cancer, chronic pancreatitis, intraductal papillary mucinous neoplasms (IPMN), and age-matched controls. A total of 124 patients, including PDAC (N=93), chronic pancreatitis of different etiologies (N=12), IPMN (N=8), and controls (N=11), were included in the discovery cohort. PDAC diagnosis was established by histology or cytology, and staging was performed according to the American Joint Committee on Cancer guidelines (8th Edition 2016; *Amin et al., 2017*; *Supplementary file 1*). For the independent validation cohort, a total of 36 patients were enrolled, including PDAC (N=24), IPMN (N=6), and chronic pancreatitis (N=6; *Supplementary file 9*).

## Plasma sample collection and processing

All blood samples were collected and processed following the same standard operating procedure optimized for EV analysis and included the following steps: (i) whole blood was collected into one 10 ml yellow-top tube containing acid citrate dextrose; (ii) blood was mixed by gently inverting the tube five times; (iii) vacutainer tubes were stored upright at room temperature (RT); (iv) samples were centrifuged at 1300 × $g$ for 15 min in RT; (v) plasma was removed from the top carefully avoiding cell pellet; (vi) repeat centrifugation of plasma at 2500 × $g$ for 15 min in RT; (vii) again, plasma was removed from the top carefully avoiding cell pellet; (viii) third centrifugation at 2500 × $g$ for 15 min in RT, then samples were aliquoted to be stored at –80 °C.

## Extracellular vesicle isolation from plasma

We employed EVtrap for EV isolation from plasma samples (*Iliuk et al., 2020*). EVtrap beads were provided by Tymora Analytical (West Lafayette, IN) as a suspension in water and were used as previously described in more details (*Iliuk et al., 2020*; *Wu et al., 2018*). Briefly, 100 µL plasma samples were diluted 20 times in the diluent buffer, the EVtrap beads were added to the samples in a 1:2 v/v ratio, and the samples were incubated by end-over-end rotation for 30 min according to the manufacturer's instructions. After supernatant removal using a magnetic separator rack, the beads were washed with PBS, and the EVs were eluted by a 10 min incubation with 200 mM triethylamine (TEA, Millipore-Sigma). The samples were fully dried in a vacuum centrifuge.

## Preparation of EV samples

The isolated and dried EV samples were lysed to extract proteins using the phase-transfer surfactant (PTS) aided procedure. The proteins were reduced and alkylated by incubation in 10 mM TCEP and 40 mM CAA for 10 min at 95 °C. The samples were diluted fivefold with 50 mM triethylammonium bicarbonate and digested with Lys-C (Wako) at 1:100 (wt/wt) enzyme-to-protein ratio for 3 hr at 37 °C. Trypsin was added to a final 1:50 (wt/wt) enzyme-to-protein ratio for overnight digestion at 37 °C. To remove the PTS surfactants from the samples, the samples were acidified with trifluoroacetic acid (TFA) to a final concentration of 1% TFA, and ethyl acetate solution was added at a 1:1 ratio. The mixture was vortexed for 2 min and then centrifuged at 16,000 × $g$ for 2 min to obtain aqueous and organic phases. The organic phase (top layer) was removed, and the aqueous phase was collected. This step was repeated once more. The samples were dried in a vacuum centrifuge and desalted using Top-Tip C18 tips (Glygen) according to the manufacturer's instructions. The samples were dried completely in a vacuum centrifuge and stored at –80 °C.

## LC–MS analysis of plasma EV proteome

Approximate 1 µg of each dried peptide sample was dissolved in 10.5 µL of 0.05% trifluoroacetic acid with 3% (vol/vol) acetonitrile containing spiked-in indexed Retention Time Standard containing 11 artificially synthetic peptides (Biognosys). The spiked-in 11-peptides standard mixture was used to account for any variation in retention times and to normalize abundance levels among samples. 10 µL of each sample was injected into an Ultimate 3000 nano UHPLC system (Thermo Fisher Scientific). Peptides were captured on a 2 cm Acclaim PepMap trap column and separated on a heated 50 cm Acclaim PepMap column (Thermo Fisher Scientific) containing C18 resin. The mobile phase buffer consisted of 0.1% formic acid in ultrapure water (buffer A) with an eluting buffer of 0.1% formic acid in 80% (vol/vol) acetonitrile (buffer B) run with a linear 60 min gradient of 6–30% buffer B at a flow rate of 300 nL/min. The UHPLC was coupled online with a Q-Exactive HF-X mass spectrometer (Thermo Fisher Scientific). The mass spectrometer was operated in the data-dependent mode, in which a full-scan MS (from m/z 375–1500 with the resolution of 60,000) was followed by MS/MS of the 15 most intense ions (30,000 resolution; normalized collision energy - 28%; automatic gain control target (AGC) - 2E4, maximum injection time - 200ms; 60 sec exclusion).

## EV proteome data processing

The raw files were searched directly against the human Swiss-Prot database with no redundant entries using Byonic (Protein Metrics) and Sequest search engines loaded into Proteome Discoverer 2.3 software (Thermo Fisher Scientific). MS1 precursor mass tolerance was set at 10 ppm, and MS2 tolerance was set at 20ppm. Search criteria included a static carbamidomethylation of cysteines (+57.0214 Da)

and variable modifications of oxidation (+15.9949 Da) on methionine residues and acetylation (+42.011 Da) at the N terminus of proteins. The search was performed with full trypsin/P digestion and allowed a maximum of two missed cleavages on the peptides analyzed from the sequence database. The false-discovery rates of proteins and peptides were set at 0.01. All protein and peptide identifications were grouped, and any redundant entries were removed. Only unique peptides and unique master proteins were reported.

All data were quantified using the label-free quantitation node of Precursor Ions Quantifier through the Proteome Discoverer v2.3 (Thermo Fisher Scientific). For the quantification of proteomic data, the intensities of peptides were extracted with initial precursor mass tolerance set at 10 ppm, a minimum number of isotope peaks as 2, maximum $\Delta$RT of isotope pattern multiplets – 0.2 min, PSM confidence FDR of 0.01, with hypothesis test of ANOVA, maximum RT shift of 5 min, pairwise ratio-based ratio calculation, and 100 as the maximum allowed fold change. The abundance levels of all peptides and proteins were normalized to the spiked-in internal iRT standard. For calculations of fold-change between the groups of proteins, total protein abundance values were added together, and the ratios of these sums were used to compare proteins within different samples.

The abundances of EV proteins were normalized using indexed retention time (iRT) in Proteome Discoverer (Thermo Fisher Scientific). Abundances were categorized into four different categories: Control, Chronic Pancreatitis, IPMN, and PDAC. Protein abundances were then log2 transformed and quantile normalized for further analysis.

A non-parametric Wilcox Rank Sum test was performed to test the null hypothesis that the distributions of two groups of the patient population are the same, and the fold change and p-values for each protein were estimated for the following comparisons: IPMN vs. Control, CP vs. Control, PDAC vs. Control, Benign Pancreatic Diseases (CP, IPMN) vs. PDAC. Multiple testing correction was done using Benjamini-Hochberg method to control for the false discovery rate (*Benjamini and Hochberg, 1995*). Volcano plots were created using those p values and fold change. Heatmaps visualization and clustering of statistically significant proteins, with adjusted p-value ≤0.05 and absolute fold change ≥2, were created in R using the pheatmap package. Euclidean distance and average cluster method were used. The values were row-scaled for normalization. Both rows and columns were allowed to cluster.

## Pathways enrichment and protein network analysis

Pathway enrichment analysis was performed on statistically significant genes using g:Profiler (*Raudvere et al., 2019*), a web-based tool that searches for pathways whose genes are significantly enriched in our dataset compared to a collection of genes representing Gene Ontology (GO) terms and Reactome pathways. We further used EnrichmentMap (*Merico et al., 2010*), a Cytoscape, v3.8.2 (*Shannon et al., 2003*) application to create a visual network of connected pathways that helps to identify relevant pathways and theme (*Reimand et al., 2019*). A Protein-Protein interaction network was generated using a stringApp, a Cytoscape app. This application allows to import STRING networks into Cytoscape and enables to perform complex network analysis and visualization of networks (*Szklarczyk et al., 2019*).

## Parallel reaction monitoring and data analysis

Parallel reaction monitoring mass spectrometry (PRM-MS) was employed for validation experiments. Twenty-five cEV markers were selected for validation based on fold change increase ≥5.5, p-value ≤0.01, and technical aspects (number of unique peptides and coverage; *Supplementary file 8*). Thirty-six plasma samples from a new cohort were used for the validation (24 PDAC, 6 IPMN and 6 chronic pancreatitis samples). The EVs were isolated from plasma and the proteins processed as described before. Peptide samples were dissolved in 10.8 µL 0.05% TFA & 2% ACN, and 10 µL injected into the UHPLC coupled with a Q-Exactive HF-X mass spectrometer (Thermo Fisher Scientific). The mobile phase buffer consisted of 0.1% formic acid in HPLC grade water (buffer A) with an eluting buffer containing 0.1% formic acid in 80% (vol/vol) acetonitrile (buffer B) run with a linear 60 min gradient of 5–35% buffer B at a flow rate of 300 nL/min. Each sample was analyzed under PRM with an isolation width of ±0.8 Th. In these PRM experiments, an MS2 level at 30,000 resolution relative to m/z 200 (AGC target 2E5, 200ms maximum injection time) was run as triggered by a scheduled inclusion list. Higher-energy collisional dissociation was used with 28 eV normalized collision energy. PRM data were manually curated within Skyline-daily (64-bit) 20.2.1.404 (32d27b598) (*MacLean et al., 2010*).

### Identification of EV signature for pancreatic cancer diagnosis

To identify a biomarker signature demonstrating the highest discriminatory power between 'benign' and 'PDAC' diseases, we adopted a binary classification approach utilizing Support Vector Machines (SVM). Recognizing that classification models built on an extensive array of proteins may incorporate irrelevant markers, which can diminish the predictive accuracy, we started with a list of significantly differentially expressed set of 91 proteins between 'benign' and 'PDAC' patients and further employed a consensus feature selection strategy using two algorithms, 'Recursive Feature Elimination' (SVM-RFE), and 'Integrated RFE with a non-parametric Wilcox rank test (sigFeature). Subsequently, we selected the top 16 markers, the classification performance of which was subjected to testing in an independent validation cohort. The classification performance evaluation of these markers, both individually and in various combinations, involved a rigorous assessment utilizing 80% of the data for training and the remaining 20% for internal-validation. To ensure the quality of the training process, we employed five repetitions of a tenfold cross-validation approach. The optimal kernel parameters were determined through tuning across a broad range of values. Receiver operating characteristic (ROC) analysis served as the metric to gauge the performance of the classifier model. All algorithms for identifying the EV signature predictive of pancreatic cancer diagnosis were implemented in R. We used Support Vector Machine (SVM) using CRAN package, e107 (*Meyer et al., 2015*). Ranking of genes was achieved using packages 'sigFeature' and 'SVM-RFE'. An R package, 'pROC' (*Robin et al., 2011*) was used to build a receiver operating characteristic curve (ROC) and to calculate the AUC.

### Survival analysis

The prognostic value of every protein was estimated by dividing patients into two groups: group 1, patients with expression below the 25th percentile, and group 2, patients with expression values greater than 75th percentile. The Kaplan-Meier estimator was used to estimate the survival function associating survival with EV protein expression, and the log-rank test was used to compare survival curves of two groups. 'survival' R package was used for the analysis.

### Statistical analysis

All statistical analyses were performed using the statistical software R. Statistical significance was calculated by two-tailed Student's t-test or Wilcoxon rank-sum test unless specified otherwise in the figure legend. Data are expressed as mean ± SEM. A p-value <0.05 in biological experiments or FDR <0.05 after multiple comparison corrections in proteomics data analysis was considered statistically significant.

## Acknowledgements

We thank the patients and their families for their participation in this study. BB was supported in part through UM1 (CA186709-06). SDF was supported in part through the Barbara Janson and Arthur Hilsinger Pancreatology Fellowship. Institutional startup funds and UO1 (CA224193) to SKM, and seed grant from Hirschberg Foundation for Pancreatic Cancer Research to LH. We thank members of the Muthuswamy laboratory for their critical input throughout the development of this project.

## Additional information

#### Competing interests

Bruno Bockorny, Lakshmi Muthuswamy, Ling Huang, Weiguo Andy Tao, Manuel Hildago, Anton Iliuk, Senthil K Muthuswamy: Inventor on a pending patent application (US20220291222A1) for pancreatic cancer detection, based on data generated from this publication. Joseph E Grossman: Employee of Agenus Inc. Supraja Narasimhan: Employee of Deciphera Pharmaceuticals. The other authors declare that no competing interests exist.

## Funding

| Funder | Grant reference number | Author |
| --- | --- | --- |
| UM1 | CA186709-06 | Bruno Bockorny |
| Institutional startup funds, Beth Israel Deaconess Medical Center | | Senthil K Muthuswamy |
| Hirshberg Foundation | | Ling Huang |
| National Institutes of Health | UO1 (CA224193) | Senthil K Muthuswamy |

The funders had no role in study design, data collection and interpretation, or the decision to submit the work for publication.

## Author contributions

Bruno Bockorny, Conceptualization, Resources, Data curation, Formal analysis, Funding acquisition, Validation, Investigation, Visualization, Methodology, Writing – original draft, Project administration, Writing – review and editing; Lakshmi Muthuswamy, Data curation, Software, Formal analysis, Investigation, Visualization, Methodology, Writing – original draft, Writing – review and editing; Ling Huang, Data curation, Formal analysis, Investigation, Visualization, Methodology, Writing – original draft, Writing – review and editing; Marco Hadisurya, Data curation, Formal analysis, Writing – review and editing; Christine Maria Lim, Data curation, Investigation, Visualization, Writing – review and editing; Leo L Tsai, Ritu R Gill, Jesse L Wei, Joseph E Grossman, Robert J Besaw, Mandeep S Sawhney, Steven D Freedman, Resources, Investigation, Writing – review and editing; Andrea J Bullock, Resources; Supraja Narasimhan, Sofia Perea, Project administration, Writing – review and editing; Weiguo Andy Tao, Resources, Investigation, Methodology, Writing – review and editing; Manuel Hildago, Resources, Supervision, Investigation, Writing – original draft, Writing – review and editing; Anton Iliuk, Conceptualization, Resources, Data curation, Formal analysis, Supervision, Investigation, Methodology, Writing – original draft, Writing – review and editing; Senthil K Muthuswamy, Conceptualization, Resources, Supervision, Investigation, Writing – original draft, Writing – review and editing

## Author ORCIDs

Bruno Bockorny ⓘ https://orcid.org/0000-0002-9162-1560
Ling Huang ⓘ https://orcid.org/0000-0001-8855-788X
Weiguo Andy Tao ⓘ https://orcid.org/0000-0002-5535-5517
Anton Iliuk ⓘ https://orcid.org/0000-0002-2914-1363
Senthil K Muthuswamy ⓘ https://orcid.org/0000-0001-6564-9634

## Ethics

We conducted this study at Beth Israel Deaconess Medical Center with the approval of the Harvard Cancer Center Institutional Review Board. All subjects provided written informed consent (DF/HCC IRB#17-640).

Reviewer #1 (Public review): https://doi.org/10.7554/eLife.87369.3.sa1
Reviewer #2 (Public review): https://doi.org/10.7554/eLife.87369.3.sa2
Author response https://doi.org/10.7554/eLife.87369.3.sa3

# Additional files

## Supplementary files

• Supplementary file 1. Baseline characteristics of patients enrolled on the discovery cohort.

• Supplementary file 2. Plasma EV analysis reproducibility.

• Supplementary file 3. LC-MS results of EV analysis of plasma from patients with PDAC (PA), IPMN, Chronic Pancreatitis (CP) and Control individuals.

• Supplementary file 4. List of EV proteins that met the eligibility criteria for principal component analysis.

- Supplementary file 5. List of 182 proteins differentially expressed in PDAC compared to benign diseases.
- Supplementary file 6. List of EV proteins that are significantly altered in patients with metastatic versus non-metastatic diseases.
- Supplementary file 7. Table A: Support Vector Machine Prediction model output for the 16 individual markers included in the in External Validation Cohorts. Table B: The contingency table for 7-biomarker signature, offering insights into model accuracy for both the Internal-Discovery and External Validation cohorts.
- Supplementary file 8. List of 25 cEV proteins that met the eligibility criteria for validation studies.
- Supplementary file 9. Baseline characteristics of patients enrolled in the validation cohort.
- MDAR checklist

### Data availability

All identified proteins are included in *Supplementary files 1–9*.

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
