## [Editor Report · eLife Assessment]

The authors analyze a comprehensive cohort of human plasma samples to identify an extracellular vesicles protein signature for early diagnosis of pancreatic cancer. The application of liquid biopsies is **valuable**, and the work addresses a key clinical problem as pancreas cancer is often diagnosed in late stages. The strength of evidence is **solid**. Altogether, this work supports the potential use of extracellular vesicles in clinical settings, with promising value to scientists and clinicians.

---

## [Referee Report · Reviewer #1 (Public review)]

This study presents a large cohort of plasma-derived extracellular vesicle samples from 124 individuals, including patients with PDAC, benign pancreatic diseases and controls. The authors identified a panel of protein markers for the early detection of pancreatic cancer and validated in an external cohort.

---

## [Referee Report · Reviewer #2 (Public review)]

This work investigates the use of extracellular vesicles (EVs) in blood as a noninvasive 'liquid biopsy' to aid in differentiation of patients with pancreatic cancer (PDAC) from those with benign pancreatic disease and healthy controls, an important clinical question where biopsies are frequently non-diagnostic. The use of extracellular vesicles as biomarkers of disease has been gaining interest in recent history, with a variety of published methods and techniques, looking at a variety of different compositions ('the molecular cargo') of EVs particularly in cancer diagnosis (Shah R, et al, N Engl J Med 2018; 379:958-966).

This study adds to the growing body of evidence in using EVs for earlier detection of pancreatic cancer, identifying both new and known proteins of interest. Limitations in studying EVs in general include dealing with low concentrations in circulation and identifying the most relevant molecular cargo. This study provides validation of assaying EVs using the novel EVtrap method (Extracellular Vesicles Total Recovery And Purification), which the authors show to be more efficient than current standard techniques and potentially more scalable for larger clinical studies.

The strength of this study is in its numbers - the authors worked with a cohort of 124 cases, 93 of them which were PDAC samples, which considered large for an EV study (Jia, E et al. BMC Cancer 22, 573 (2022)). The benign disease group (n=20, between chronic pancreatitis and IPMNs) and healthy control groups (n=11) were relatively small, but the authors were not only able to identify candidate biomarkers for diagnosis that clearly stood out in the PDAC cohort, but also validate it in an independent cohort of 36 new subjects. Proteins they've identified as associated with pancreatic cancer over benign disease included PDCD6IP, SERPINA12 and RUVBL2. They were even able to identify a set of EV proteins associated with metastasis and poorer prognosis , which include the proteins PSMB4, RUVBL2 and ANKAR and CRP, RALB and CD55. Their 7-EV protein signature yielded an 89% prediction accuracy for the diagnosis of PDAC against a background of benign pancreatic diseases that is compelling and comparable to other studies in the literature (Jia, E. et al. BMC Cancer 22, 573 (2022)).

The limitations of this study are its containment within a single institution - further studies are warranted to apply the authors' 7-EV protein PRAC panel to multiple other cases at other institutions in a larger cohort.

---

## [Author Response]

Author response:

The following is the authors’ response to the original reviews.

**Reviewer #1 (Public Review):**
In this manuscript, Bockorny, Muthuswamy, and Huang et al. performed proteomics analysis of plasma extracellular vesicles (EVs) from pancreatic ductal adenocarcinoma (PDAC) patients and patients with benign pancreatic diseases (chronic pancreatitis and intraductal papillary mucinous neoplasm, IPMN) to develop a 7-EV protein signature that predicts PDAC. Moreover, the authors identified PSMB4, RUVBL2, and ANKAR as being associated with metastasis. These studies provide important insight into alterations of EVs during PDAC progression and the data supporting predict PDAC with EV protein signatures are solid. However, there are certain concerns regarding the rigor and novelty of the data analysis and interpretation, as well as the clinical implications, as detailed below.(1) Plasma EVs were characterized by transmission electron microscopy and nanoparticle tracking analysis to confirm their morphology and size. The authors should also include an analysis of putative EV markers (e.g., tetraspanins, syntenin, ALIX, etc.) to confirm that the analyzed particles are EVs.

We thank the reviewer for this comment. In the previous study from our co-authors who developed EVtrap method (PMID:32396726), they used electron microscopy and NTA , as well as quantification of typical EV protein markers, such as CD9, to confirm that particles isolated using EVtrap had typical characteristics of the extracellular vesicles. As such, these experiments were not replicated here. We added the following statement to the manuscript:

“Previous analyses using electron microscopy and nanoparticle tracking also confirmed that the vast majority of particles isolated by EVtrap had diameters between 100-200 nm, consistent with exosomes (PMID:32396726). In addition, EVtrap isolates demonstrates higher abundance of CD9, a common exosome marker, as compared to isolates from other traditional EV isolation methods such as size exclusion chromatography and ultracentrifugation (PMID:32396726)”

(2) The authors identified multiple over-expressed proteins in PDAC based on their foldchange and p-value; however, due to the heterogeneity of PDAC, it is necessary to show a heatmap displaying their abundance in all samples. High fold change does not necessarily indicate consistently high abundance in all PDAC samples.

We thank the reviewer for this suggestion. We have now included the heatmap in the new Supplementary Figure 3.

(3) PSMB4, RUVBL2, and ANKAR were identified as being associated with metastasis. The authors state that they intended to distinguish early and late-stage cancer samples, but it is unclear why they chose to compare metastatic and non-metastatic samples, as the non-metastatic group also includes late-stage cancer samples. This sentence should be rephrased to more accurately reflect the sample types profiled.

We thank the reviewer for pointing this out. We would like to clarify that this analyses shown in Figures 3B and 3C pertain to patients with Metastatic vs Non-Metastatic disease, not early versus late stage. We edited the text to ensure this information is clear.

(4) Non-metastatic and metastatic patients were separated based on global protein abundance. The samples within each group display significant heterogeneity, with some samples displaying similar patterns although they were classified into different groups (Figure 3A), and the samples within the same group, particularly the metastasis group, did not consistently exhibit similar patterns of protein abundance. The authors should clarify this point.

We thank the reviewer for this comment. The EV proteomic expression is anticipated not to show the exact pattern across of samples of each group. The purpose of this experiment depicted in Figure 3 heatmap is to show the enrichment for pattern of expressions, but we acknowledge that not all samples from the same group have the exact proteome pattern.

We added this statement in the discussion section:

“As expected, the EV proteomic profiles of PDAC patients exhibited significant heterogeneity. While the above mentioned markers exhibited strong association with disease states at population levels, their abundances in individual patients varied significantly. Those observations highlight the need to develop multi-protein panels for pancreatic cancer diagnosis and prognosis.”

(5) The authors performed the survival analysis on a set of EV proteins but did not specify the origin of these markers or how many markers were examined. The authors should show their abundances across different groups, such as different stages and metastasis status.

We thank the reviewer for the comments. The goal of this experiment was not to identify EV proteins that performed similarly well for diagnosing and prognostication. In Figure 3A, 3B and 3C, we identified EV proteins that had better performance for diagnosis of metastatic disease. In these experiments we made comparative analysis between patients with metastasis versus non-metastasis. In the experiment depicted in Figure 3D, the goal was to identify EV markers that had better performance is prognosticating outcomes as measured by overall survival, out of the markers identified in the previous experiments from Figure 3A. We would like to further clarify that based on our observation and others, it has become clear that EV profiles from cancer patients are highly heterogenous and we do not anticipate that a single marker will have sufficient test performance for cancer diagnosis or prognosis assessment when measured isolated. Rather, we anticipate that a panel of markers may yield better performance for diagnosis while a different combination of EV markers may have better performance for prognosis assessment.

(6) The classification model yielded a 100% accuracy, which may refer to AUC, in their discovery cohort, but it decreased to 89% in the independent cohort. This suggests that the authors have encountered overfitting issues with their model, where it performed well on the discovery cohort but did not generalize well to the independent cohort. The authors should clarify this point. The AUC score of the 7-EV signature is 0.89 and is not equivalent to prediction accuracy. In order to demonstrate prediction accuracy, the authors should show the confusion matrix of training and testing data as well as other evaluation metrics, such as accuracy, precision, and recall.

We thank the reviewer for providing these insightful comments. As you noted, the 7-biomarker signature machine learning model attained an impressive 100% accuracy within the internal Discovery Cohort, raising concerns about potential overfitting in the external validation dataset. Acknowledging the noted difference in AUROC of 0.11 in the external validation cohort, which surpasses the typical reported range of ~0.06-0.09, the model demonstrated a commendable AUROC of 0.89 in an independent patient cohort. Moreover, the utilization of an alternate technology to measure protein abundance in the validation dataset, underscores the model’s reproducibility and validity. We have provided the model metrics for both internal- and external-validation cohort. For these, please see updated Supplementary Figure 7, as well as the new Supplementary Figure 6 and Supplementary Figure 8. We also amended the discussion section to acknowledge that the validation cohort had limited sample size and proteins were measured in using a different method. Those factors likely contributed to the lower accuracy of predictions in the validation cohort. We addressed these limitations in the discussion section of the manuscript.

(7) The authors should include more details of their model and the process of selection of signatures to enhance the reproducibility and transparency of their methods.

We thank the reviewer for their valuable comments. To enhance clarity, we have incorporated additional information regarding the method employed for biomarker signature identification into the ‘Methods Section’ in page 23. We note that Supplementary Table 7a provides details on ‘Sensitivity, Specificity, Precision, and AUC’ for the 16 markers included in the external validation study. Additionally, Supplementary Table 7b presents the contingency table for 7-biomarker signature, offering insights into model accuracy for both the Internal-Discovery and External Validation cohorts.

**Reviewer #2 (Public Review):**
The authors intended to identify a protein signature in extracellular vesicles of serum to distinguish pancreatic ductal adenocarcinoma from benign pancreatic diseases.A major strength of the work presented is the valuable profiling of a significant number of patient samples, with a rich cohort of patients with pancreatic cancer, benign pancreatic diseases, and healthy controls. However, despite the strong cohorts presented, the numbers of patient samples for benign pancreatic diseases as well as controls were very limited.Also, the method used to isolate vesicles, EVTrap, recognizes double bilayers, which means that it can detect cellular debris and apoptotic bodies, which are very common in the circulation of patients that are undergoing chemotherapy. It would be important to identify the patients that are therapy naïve and the ones that are not because of this possible bias.

We thank the Reviewer for these comments. We want to point out that the experiments presented in Supplementary Figure 1 (Transmission electron microscopy images and Nanoparticle tracking analysis) confirm that the vesicles isolated with EVTrap are not cellular debris and apoptotic bodies. Rather, these structures are in the nano range expected for exosomes. This is further supported by the additional work from our co-author and collaborator describing the development of EVtrap and its performance in isolating exosomes when compared to other traditional methods such as ultracentrifugation and size exclusion chromatography (PMID:32396726).

As per the Reviewer’s request, we have provided an additional heatmap figure depicting whose patients are treatment naïve to differentiate from those who have received treatment (revised Figure 2C).

Additionally, the transmission electron microscopy data reflect this heterogeneity of the samples, also with little identification of double bilayered vesicles. It would be important to identify some extracellular vesicles markers in those preparations to strengthen the quality of the samples analyzed.

We appreciate the comment from the Reviewer and acknowledge the importance of identifying exosome markers on the isolate from EVtrap. These experiments have already been done and are reported in the original paper describing the development of this method by our co-authors in a separate work. In the manuscript PMID: 30080416, our collaborators demonstrated the detection of CD9, a well-known exosome marker, using Western Blot from isolates using EVtrap or ultra-centrifugation, a traditional technique to isolate exosomes. This work showed that EVtrap yielded much higher recovery rate of exosomes with lower contamination from soluble proteins. We did not repeat these already published experiments, but we amended our manuscript to reference these results.

What is more, previously published work with this same methodology identifies around 2000 proteins per sample. It would be important to explain why in this study there seems to be a reduction in more than 50% of the amount of proteins identified in the vesicles.

We thank the Reviewer for pointing out this important detail. In the previous work in which EVtrap was developed by our co-authors, the blood samples were processed using a different protocol, with shorter centrifugation (2,500g for 10 min) (PMID: 32396726). In the current work, we employed three centrifugation steps. As detailed in the Methods section of the manuscript, blood samples were centrifuged at 1,300g for 15 min. Then plasma was removed from the top carefully avoiding cell pellet; Repeat centrifugation of plasma at 2,500g for 15 min; Again, plasma was removed from the top carefully avoiding cell pellet; Third centrifugation at 2,500g for 15 min. This more extensive centrifugation process was intended to further increase the removal of platelets, apoptotic bodies, and other large particles and aggregates. Accordingly, we anticipate that the additional centrifugation steps decreased the contamination of our isolates but may have also decreased the amount of exosome proteins, hence the lower amount of exosome proteins identified in our study as compared to the original study from our co-authors (PMID: 32396726).

One of the proteins that constantly surges on the analysis is KRT20. It would be important to proceed with the analysis by first filtering out possible contaminants of the proteomics, of which keratins are the most common ones.

We thank the Reviewer for this comment. We would like to point out that we do believe that KRT20 is, in fact, cancer related and a not a contaminant. This is supported by our results presented in this manuscript showing enrichment or KRT20 in PDAC cases, and lower expression in benign samples. If this protein was a contaminant, its expression would be found uniformly in all samples, there would be no apparent reason for different expression between malignant vs benign cases, as all samples were processed following the same procedures. In addition, increased expression of KRT20 in PDAC tissues has also been reported by others. For instance, in a study by Schmiz-Winnthal (PMID: 16364723), the authors showed that Cytokeratin 20 (KRT20) were expressed in 76% of PDAC patients and expression of KRT20 was associated with poor survival after surgical resection. Based on these observations, we believe that the KRT20 identified in our study is indeed a tumor associated EV protein rather than contamination.

Finally, none of the 7-extracellular vesicle protein signatures has been validated by other techniques, such as western blot, in extracellular vesicles isolated by other, standard, methods, such as size exclusion chromatography.A distinct technique for protein analysis was done but not a different method of isolation of these vesicles. This would strengthen the results and the origin of the proteins.

We appreciate the Reviewer’s comment. We would like to again emphasize that the goal of this manuscript was not to compare the performance of EVtrap with other traditional EV isolation approaches such as ultracentrifugation and size exclusion chromatography. The main goal of study is to determine proteomic profiles of EVs isolated from clinical samples and provide such information to research community for further studies. As the Reviewer points out, proteins in EVs are highly heterogeneous which highlight the complexity of EV biology and interpatient heterogeneity of pancreatic cancer. We do not anticipate the development of EV-based markers for pancreatic diagnosis can be achieved by a single team, but by a community of researchers. We hope information presented in the current study will help other researchers identify additional candidates for validation in future work. Nonetheless, we edited the manuscript to discuss the limitation of not doing cross-validation of protein detection using a different method.

The conclusions that are reached do not fully meet the proposed aims of the identification of a protein signature in circulating extracellular vesicles that could improve early detection of the disease. The authors did not demonstrate the superiority of detection of these proteins in extracellular vesicles versus simply performing an ELISA, nor their superiority with respect to the current standard procedure for diagnosis.

We would like to clarify to the Reviewer that the goal of this manuscript was not to prove superiority of the EV signature biomarker in diagnosing pancreatic cancer as compared to current standard of care (SOC) practice, i.e., CT scans, endoscopic ultrasound and CA19-9. In order to prove such superiority, one would require a large, randomized phase III trial with several hundred patients. This was not the pursue of our discovery EV proteomics study and we double checked our manuscript to ensure no such claim was made. Rather, we aimed at developing a new pipeline for discovery of new EV biomarkers and we believe we were able to prove that this approach was successful in discovering a new class of biomarkers based on proteins expressed on extra-cellular vesicles that have predominant expression on patients with pancreatic cancer. Future studies should continue to advance this field with goals of improving on the current standard of care diagnostic methods.

The authors also suggest that profiling of circulating extracellular vesicles provides unique insights into systemic immune changes during pancreatic cancer development. How is this better than a regular hemogram is not clear.

We would like to clarify that the overall goal of this study is to provide patient-relevant information for the research community to further investigate biology of extracellular vesicles. For the state 'unique insights into systemic immune changes' we referred to the fact that we discovered EVs carrying proteins involved in immune responses. Previous studies have shown that EVs play important roles in cell-cell communication, discoveries from our study provide candidates for future studies on cellular mechanisms underlying immune regulation during pancreatic cancer development.

Finally, it would be important to determine how this signature compares with many others described in the literature that have the exact same aim. Why and how would this one be better?

We would like to again clarify that comparing the diagnostic performance of the EV biomarkers discovered in the study against standard of care methods (CA19-9, ctDNA, CT scan) was beyond the scope of this discovery EV proteomics work. We reviewed the manuscript to ensure that no claims were made as far as superiority against point-of-care tests available in clinic.

**Reviewer #3 (Public Review):**
This work investigates the use of extracellular vesicles (EVs) in blood as a noninvasive 'liquid biopsy' to aid in the differentiation of patients with pancreatic cancer (PDAC) from those with benign pancreatic disease and healthy controls, an important clinical question where biopsies are frequently non-diagnostic. The use of extracellular vesicles as biomarkers of disease has been gaining interest in recent history, with a variety of published methods and techniques, looking at a variety of different compositions ('the molecular cargo') of EVs particularly in cancer diagnosis (Shah R, et al, N Engl J Med 2018; 379:958-966).This study adds to the growing body of evidence in using EVs for earlier detection of pancreatic cancer, identifying both new and known proteins of interest. Limitations in studying EVs, in general, include dealing with low concentrations in circulation and identifying the most relevant molecular cargo. This study provides validation of assaying EVs using the novel EVtrap method (Extracellular Vesicles Total Recovery And Purification),which the authors show to be more efficient than current standard techniques and potentially more scalable for larger clinical studies.The strength of this study is in its numbers - the authors worked with a cohort of 124 cases,93 of them which were PDAC samples, which are considered large for an EV study (Jia, E etal. BMC Cancer 22, 573 (2022)). The benign disease group (n=20, between chronic pancreatitis and IPMNs) and healthy control groups (n=11) were relatively small, but the authors were not only able to identify candidate biomarkers for diagnosis that clearly stood out in the PDAC cohort, but also validate it in an independent cohort of 36 new subjects.Proteins they have identified as associated with pancreatic cancer over benign disease included PDCD6IP, SERPINA12, and RUVBL2. They were even able to identify a set of EV proteins associated with metastasis and poorer prognosis, which include the proteins PSMB4, RUVBL2 and ANKAR and CRP, RALB and CD55. Their 7-EV protein signature yielded an 89% prediction accuracy for the diagnosis of PDAC against a background of benign pancreatic diseases that is compelling and comparable to other studies in the literature (Jia,E. et al. BMC Cancer 22, 573 (2022)).The limitations of this study are its containment within a single institution - further studies are warranted to apply the authors' 7-EV protein PRAC panel to multiple other cases at other institutions in a larger cohort.

We are very thankful to the Reviewer for the positive feedback. We are similarly optimistic that EV-based biomarkers will assist future researchers to develop better diagnostic assays for patients with pancreatic cancer, as well as other tumor types lacking accurate blood-based tests.